# Evolved Policy Gradients

**Rein Houthooft**[*], **Richard Y. Chen**[*], **Phillip Isola**[*†×], **Bradly C. Stadie**[*†], **Filip Wolski**[*],
**Jonathan Ho**[*†], **Pieter Abbeel**[†]
OpenAI[*], UC Berkeley[†], MIT[×]

## Abstract

We propose a metalearning approach for learning gradient-based reinforcement learning (RL) algorithms. The idea is to evolve a differentiable loss function, such that an agent, which optimizes its policy to minimize this loss, will achieve high rewards. The loss is parametrized via temporal convolutions over the agent's experience. Because this loss is highly flexible in its ability to take into account the agent's history, it enables fast task learning. Empirical results show that our evolved policy gradient algorithm (EPG) achieves faster learning on several randomized environments compared to an off-the-shelf policy gradient method. We also demonstrate that EPG's learned loss can generalize to out-of-distribution test time tasks, and exhibits qualitatively different behavior from other popular metalearning algorithms.

## 1   Introduction

Most current reinforcement learning (RL) agents approach each new task de novo. Initially, they have no notion of what actions to try out, nor which outcomes are desirable. Instead, they rely entirely on external reward signals to guide their initial behavior. Coming from such a blank slate, it is no surprise that RL agents take far longer than humans to learn simple skills [12].

Our aim in this paper is to devise agents that have a prior notion of what constitutes making progress on a novel task. Rather than encoding this knowledge explicitly through a learned behavioral policy, we encode it implicitly through a learned loss function. The end goal is agents that can use this loss function to learn quickly on a novel task. This approach can be seen as a form of metalearning, in which we learn a learning algorithm. Rather than mining rules that generalize across data points, as in traditional machine learning, metalearning concerns itself with devising algorithms that generalize across tasks, by infusing prior knowledge of the task distribution [7].

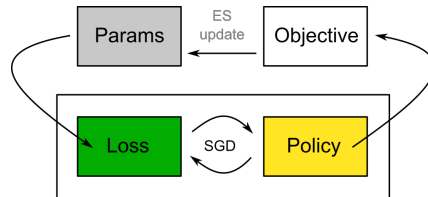

Figure 1: High-level overview of our approach.

Our method consists of two optimization loops. In the inner loop, an agent learns to solve a task, sampled from a particular distribution over a family of tasks. The agent learns to solve this task by minimizing a loss function provided by the outer loop. In the outer loop, the parameters of the loss function are adjusted so as to maximize the final returns achieved after inner loop learning. Figure 1 provides a high-level overview of this approach.

Although the inner loop can be optimized with stochastic gradient descent (SGD), optimizing the outer loop presents substantial difficulty. Each evaluation of the outer objective requires training a complete inner-loop agent, and this objective cannot be written as an explicit function of the loss

parameters we are optimizing over. Due to the lack of easily exploitable structure in this optimization problem, we turn to evolution strategies (ES) [20, 27, 9, 21] as a blackbox optimizer. The evolved loss $L$ can be viewed as a surrogate loss [24, 25] whose gradient is used to update the policy, which is similar in spirit to policy gradients, lending the name "evolved policy gradients".

The learned loss offers several advantages compared to current RL methods. Since RL methods optimize for short-term returns instead of accounting for the complete learning process, they may get stuck in local minima and fail to explore the full search space. Prior works add auxiliary reward terms that emphasize exploration [3, 10, 17, 32, 2, 18] and entropy loss terms [16, 23, 8, 14]. Using ES to evolve the loss function allows us to optimize the true objective, namely the final trained policy performance, rather than short-term returns, with the learned loss incentivizing the necessary exploration to achieve this. Our method also improves on standard RL algorithms by allowing the loss function to be adaptive to the environment and agent history, leading to faster learning and the potential for learning without external rewards.

There has been a flurry of recent work on metalearning policies, e.g., [5, 33, 6, 13], and it is worth asking why metalearn the loss as opposed to directly metalearning the policy? Our motivation is that we expect loss functions to be the kind of object that may generalize very well across substantially different tasks. This is certainly true of hand-engineered loss functions: a well-designed RL loss function, such as that in [26], can be very generically applicable, finding use in problems ranging from playing Atari games to controlling robots [26]. In Section 4.3, we find evidence that a loss learned by EPG can train an agent to solve a task *outside the distribution* of tasks on which EPG was trained. This generalization behavior differs qualitatively from MAML [6] and RL$^2$ [5], methods that directly metalearn policies.

Our contributions include the following: 1) Formulating a metalearning approach that learns a differentiable loss function for RL agents, called EPG; 2) Optimizing the parameters of this loss function via ES, overcoming the challenge that final returns are not explicit functions of the loss parameters; 3) Designing a loss architecture that takes into account agent history via temporal convolutions; 4) Demonstrating that EPG produces a learned loss that can train agents faster than an off-the-shelf policy gradient method; 5) Showing that EPG's learned loss can generalize to *out-of-distribution* test time tasks, exhibiting qualitatively different behavior from other popular metalearning algorithms. An implementation of EPG is available at `http://github.com/openai/EPG`.

## 2   Notation and Background

We model reinforcement learning [30] as a Markov decision process (MDP), defined as the tuple $\mathcal{M} = (\mathcal{S}, \mathcal{A}, T, R, p_0, \gamma)$, where $\mathcal{S}$ and $\mathcal{A}$ are the state and action space. The transition dynamic $T : \mathcal{S} \times \mathcal{A} \times \mathcal{S} \mapsto \mathbb{R}_+$ determines the distribution of the next state $s_{t+1}$ given the current state $s_t$ and the action $a_t$. $R : \mathcal{S} \times \mathcal{A} \mapsto \mathbb{R}$ is the reward function and $\gamma \in (0, 1)$ is a discount factor. $p_0$ is the distribution of the initial state $s_0$. An agent's policy $\pi : \mathcal{S} \mapsto \mathcal{A}$ generates an action after observing a state. An episode $\tau \sim \mathcal{M}$ with horizon $H$ is a sequence $(s_0, a_0, r_0, \ldots, s_H, a_H, r_H)$ of state, action, and reward at each timestep $t$. The discounted episodic return of $\tau$ is defined as $R_\tau = \sum_{t=0}^{H} \gamma^t r_t$, which depends on the initial state distribution $p_0$, the agent's policy $\pi$, and the transition distribution $T$. The expected episodic return given agent's policy $\pi$ is $\mathbb{E}_\pi[R_\tau]$. The optimal policy $\pi^*$ maximizes the expected episodic return $\pi^* = \arg\max_\pi \mathbb{E}_{\tau \sim \mathcal{M}, \pi}[R_\tau]$. In high-dimensional reinforcement learning settings, the policy $\pi$ is often parametrized using a deep neural network $\pi_{\boldsymbol{\theta}}$ with parameters $\boldsymbol{\theta}$. The goal is to solve for $\boldsymbol{\theta}^*$ that attains the highest expected episodic return

$$\boldsymbol{\theta}^* = \arg\max_{\boldsymbol{\theta}} \mathbb{E}_{\tau \sim \mathcal{M}, \pi_{\boldsymbol{\theta}}}[R_\tau]. \tag{1}$$

This objective can be optimized via policy gradient methods [34, 31] by stepping in the direction of $\mathbb{E}[R_\tau \nabla \log \pi(\tau)]$. This gradient can be transformed into a surrogate loss function [24, 25]

$$L_{\mathrm{pg}} = \mathbb{E}[R_\tau \log \pi(\tau)] = \mathbb{E}\left[R_\tau \sum_{t=0}^{H} \log \pi(a_t|s_t)\right], \tag{2}$$

such that the gradient of $L_{\mathrm{pg}}$ equals the policy gradient. This loss function is oftent transformed through variance reduction techniques including actor-critic algorithms [11]. However, this procedure

remains limited since it relies on a particular form of discounting returns, and taking a fixed gradient step with respect to the policy. Our approach instead learns a loss. Thus, it may be able to discover more effective surrogates for making fast progress toward the ultimate objective of maximizing final returns.

## 3 Methodology

We aim to learn a loss function $L_\phi$ that outperforms the usual policy gradient surrogate loss [24]. The learned loss function consists of temporal convolutions over the agent's recent history. In addition to internalizing environment rewards, this loss could, in principle, have several other positive effects. For example, by examining the agent's history, the loss could incentivize desirable extended behaviors, such as exploration. Further, the loss could perform a form of system identification, inferring environment parameters and adapting how it guides the agent as a function of these parameters (e.g., by adjusting the effective learning rate of the agent). The loss function parameters $\phi$ are evolved through ES and the loss trains an agent's policy $\pi_\theta$ in an on-policy fashion via stochastic gradient descent.

### 3.1 Metalearning Objective

We assume access to a distribution $p(\mathcal{M})$ over MDPs. Given a sampled MDP $\mathcal{M}$, the inner loop optimization problem is to minimize the loss $L_\phi$ with respect to the agent's policy $\pi_\theta$:

$$\boldsymbol{\theta}^* = \arg\min_{\boldsymbol{\theta}} \mathbb{E}_{\tau \sim \mathcal{M}, \pi_{\boldsymbol{\theta}}}[L_\phi(\pi_{\boldsymbol{\theta}}, \tau)]. \tag{3}$$

Note that this is similar to the usual RL objectives (Eqs. (1) (2)), except that we are optimizing a learned loss $L_\phi$ rather than directly optimizing the expected episodic return $\mathbb{E}_{\mathcal{M}, \pi_{\boldsymbol{\theta}}}[R_\tau]$ or other surrogate losses. The outer loop objective is to learn $L_\phi$ such that an agent's policy $\pi_{\boldsymbol{\theta}^*}$ trained with the loss function achieves high expected returns in the MDP distribution:

$$\phi^* = \arg\max_{\phi} \mathbb{E}_{\mathcal{M} \sim p(\mathcal{M})} \mathbb{E}_{\tau \sim \mathcal{M}, \pi_{\boldsymbol{\theta}^*}}[R_\tau]. \tag{4}$$

### 3.2 Algorithm

The final episodic return $R_\tau$ of a trained policy $\pi_{\boldsymbol{\theta}^*}$ cannot be represented as an explicit function of the loss function $L_\phi$. Thus we cannot use gradient-based methods to directly solve Eq. (4). Our approach, summarized in Algorithm 1, relies on evolution strategies (ES) to optimize the loss function in the outer loop.

As described by Salimans et al. [21], ES computes the gradient of a function $F(\phi)$ according to $\nabla_\phi \mathbb{E}_{\boldsymbol{\epsilon} \sim \mathcal{N}(0,I)} F(\phi + \sigma\boldsymbol{\epsilon}) = \frac{1}{\sigma} \mathbb{E}_{\boldsymbol{\epsilon} \sim \mathcal{N}(0,I)} F(\phi + \sigma\boldsymbol{\epsilon})\boldsymbol{\epsilon}$. Similar formulations also appear in prior works including [29, 28, 15]. In our case, $F(\phi) = \mathbb{E}_{\mathcal{M} \sim p(\mathcal{M})} \mathbb{E}_{\tau \sim \mathcal{M}, \pi_{\boldsymbol{\theta}^*}}[R_\tau]$ (Eq. (4)). Note that the dependence on $\phi$ comes through $\boldsymbol{\theta}^*$ (Eq. (3)).

Step by step, the algorithm works as follows. At the start of each epoch in the outer loop, for $W$ inner-loop workers, we generate $V$ standard multivariate normal vectors $\boldsymbol{\epsilon}_v \in \mathcal{N}(0, \mathrm{I})$ with the same dimension as the loss function parameter $\phi$, assigned to $V$ sets of $W/V$ workers. As such, for the $w$-th worker, the outer loop assigns the $\lceil wV/W \rceil$-th perturbed loss function $L_w = L_{\phi + \sigma\boldsymbol{\epsilon}_v}$ where $v = \lceil wV/W \rceil$ with perturbed parameters $\phi + \sigma\boldsymbol{\epsilon}_v$ and $\sigma$ as the standard deviation.

Given a loss function $L_w, w \in \{1, \ldots, W\}$, from the outer loop, each inner-loop worker $w$ samples a random MDP from the task distribution, $\mathcal{M}_w \sim p(\mathcal{M})$. The worker then trains a policy $\pi_{\boldsymbol{\theta}}$ in $\mathcal{M}_w$ over $U$ steps of experience. Whenever a termination signal is reached, the environment resets with state $s_0$ sampled from the initial state distribution $p_0(\mathcal{M}_w)$. Every $M$ steps the policy is updated through SGD on the loss function $L_w$, using minibatches sampled from the steps $t - M, \ldots, t$:

$$\boldsymbol{\theta} \leftarrow \boldsymbol{\theta} - \delta_{\mathrm{in}} \cdot \nabla_{\boldsymbol{\theta}} L_w\big(\pi_{\boldsymbol{\theta}}, \tau_{t-M,\ldots,t}\big). \tag{5}$$

**Algorithm 1:** Evolved Policy Gradients (EPG)

---

1  **[Outer Loop] for** epoch $e = 1, \ldots, E$ **do**
2     Sample $\boldsymbol{\epsilon}_v \sim \mathcal{N}(\mathbf{0}, \mathrm{I})$ and calculate the loss parameter $\boldsymbol{\phi} + \sigma \boldsymbol{\epsilon}_v$ for $v = 1, \ldots, V$
3     Each worker $w = 1, \ldots, W$ gets assigned noise vector $\lceil wV/W \rceil$ as $\boldsymbol{\epsilon}_w$
4     **for** each worker $w = 1, \ldots, W$ **do**
5        Sample MDP $\mathcal{M}_w \sim p(\mathcal{M})$
6        Initialize buffer with $N$ zero tuples
7        Initialize policy parameter $\boldsymbol{\theta}$ randomly
8        **[Inner Loop] for** step $t = 1, \ldots, U$ **do**
9           Sample initial state $s_t \sim p_0$ if $\mathcal{M}_w$ needs to be reset
10          Sample action $a_t \sim \pi_{\boldsymbol{\theta}}(\cdot|s_t)$
11          Take action $a_t$ in $\mathcal{M}_w$ and receive $r_t$, $s_{t+1}$, and termination flag $d_t$
12          Add tuple $(s_t, a_t, r_t, d_t)$ to buffer
13          **if** $t \bmod M = 0$ **then**
14             With loss parameter $\boldsymbol{\phi} + \sigma \boldsymbol{\epsilon}_w$, calculate losses $L_i$ for steps $i = t - M, \ldots, t$
                using buffer tuples $i - N, \ldots, i$
15             Sample minibatches mb from last $M$ steps shuffled, compute $L_{\mathrm{mb}} = \sum_{j \in \mathrm{mb}} L_j$,
                and update the policy parameter $\boldsymbol{\theta}$ and memory parameter (Eq. (5))
16        In $\mathcal{M}_w$, using trained policy $\pi_{\boldsymbol{\theta}}$, sample several trajectories and compute mean return $R_w$
17     Update the loss parameter $\boldsymbol{\phi}$ (Eq. (6))
18  **Output:** Loss $L_{\boldsymbol{\phi}}$ that trains $\pi$ from scratch according to inner loop scheme, on MDPs $\sim p(\mathcal{M})$

---

At the end of the inner-loop training, each worker returns the final return $R_w$[1] to the outer loop. The outer-loop aggregates the final returns $\{R_w\}_{w=1}^W$ from all workers and updates the loss function parameter $\boldsymbol{\phi}$ as follows:

$$\boldsymbol{\phi} \leftarrow \boldsymbol{\phi} + \delta_{\mathrm{out}} \cdot \frac{1}{V\sigma} \sum_{v=1}^{V} F(\boldsymbol{\phi} + \sigma \boldsymbol{\epsilon}_v) \boldsymbol{\epsilon}_v, \tag{6}$$

where $F(\boldsymbol{\phi} + \sigma \boldsymbol{\epsilon}_v) = \frac{R_{(v-1)*W/V+1} + \cdots + R_{v*W/V}}{W/V}$. As a result, each perturbed loss function $L_v$ is evaluated on $W/V$ randomly sampled MDPs from the task distribution using the final returns. This achieves variance reduction by preventing the outer-loop ES update from promoting loss functions that are assigned to MDPs that consistently generate higher returns. Note that the actual implementation calculates each loss function's relative rank for the ES update. Algorithm 1 outputs a learned loss function $L_{\boldsymbol{\phi}}$ after $E$ epochs of ES updates.

At test time, we evaluate the learned loss function $L_{\boldsymbol{\phi}}$ produced by Algorithm 1 on a test MDP $\mathcal{M}$ by training a policy from scratch. The test-time training schedule is the same as the inner loop of Algorithm 1 (it is described in full in the supplementary materials).

## 3.3   Architecture

The agent is parametrized using an MLP policy with observation space $\mathcal{S}$ and action space $\mathcal{A}$. The loss has a memory unit to assist learning in the inner loop. This memory unit is a single-layer neural network to which an invariable input vector of ones is fed. As such, it is essentially a layer of bias terms. Since this network has a constant input vector, we can view its weights as a very simple form of memory to which the loss can write via emitting the right gradient signals. An experience buffer stores the agent's $N$ most recent experience steps, in the form of a list of tuples $(s_t, a_t, r_t, d_t)$, with $d_t$ the trajectory termination flag. Since this buffer is limited in the number of steps it stores, the memory unit might allow the loss function to store information over a longer period of time.

The loss function $L_{\boldsymbol{\phi}}$ consists of temporal convolutional layers which generate a context vector $f_{\mathrm{context}}$, and dense layers, which output the loss. The architecture is depicted in Figure 2.

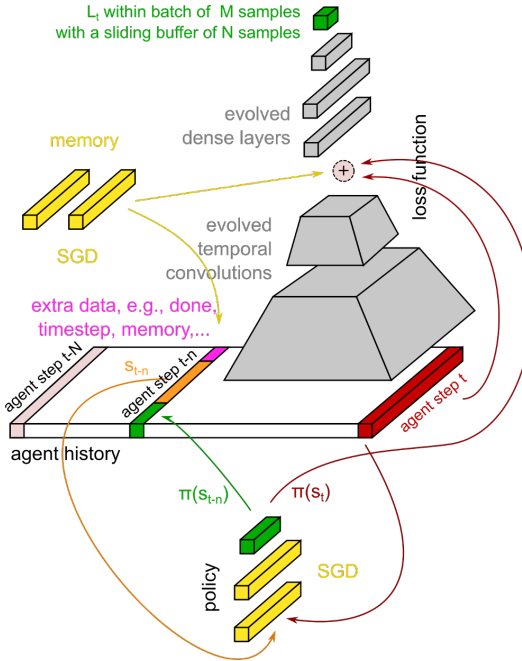

L_t within batch of M samples with a sliding buffer of N samples

memory

evolved dense layers

SGD

evolved temporal convolutions

extra data, e.g., done, timestep, memory,...

loss function

agent step t-N

s_{t-n}

agent step t-n

agent step t

agent history

$\pi(s_{t-n})$   $\pi(s_t)$

policy

SGD

At step $t$, the dense layers output the loss $L_t$ by taking a batch of $M$ sequential samples

$$\{s_i, a_i, d_i, \text{mem}, f_{\text{context}}, \pi_{\boldsymbol{\theta}}(\cdot|s_i)\}_{i=t-M}^{t}, \quad (7)$$

where $M < N$ and we augment each transition with the memory output $\text{mem}$, a context vector $f_{\text{context}}$ generated from the loss's temporal convolutional layers, and the policy distribution $\pi_{\boldsymbol{\theta}}(\cdot|s_i)$. In continuous action space, $\pi_{\boldsymbol{\theta}}$ is a Gaussian policy, i.e., $\pi_{\boldsymbol{\theta}}(\cdot|s_i) = \mathcal{N}(\cdot; \mu(s_i; \boldsymbol{\theta}_0), \Sigma)$, with $\mu(s_i; \boldsymbol{\theta}_0)$ the MLP output and $\Sigma$ a learnable parameter vector. The policy parameter vector is defined as $\boldsymbol{\theta} = [\boldsymbol{\theta}_0, \Sigma]$.

To generate the context vector, we first augment each transition in the buffer with the output of the memory unit $\text{mem}$ and the policy distribution $\pi_{\boldsymbol{\theta}}(\cdot|s_i)$ to obtain a set

$$\{s_i, a_i, d_i, \text{mem}, \pi_{\boldsymbol{\theta}}(\cdot|s_i)\}_{i=t-N}^{t}. \quad (8)$$

We stack these items sequentially into a matrix and the temporal convolutional layers take it as input and output the context vector $f_{\text{context}}$. The memory unit's parameters are updated via gradient descent at each inner-loop update (Eq. (5)).

Figure 2: Architecture of a loss computed for timestep $t$ within a batch of $M$ sequential samples (from $t - M$ to $t$), using temporal convolutions over a buffer of size $N$ (from $t - N$ to $t$), with $M \leq N$: dense net on the bottom is the policy $\pi(s)$, taking as input the observations (orange), while outputting action probabilities (green). The green block on the top represents the loss output. Gray blocks are evolved, yellow blocks are updated through SGD.

Note that both the temporal convolution layers and the dense layers do not observe the environment rewards directly. However, in cases where the reward cannot be fully inferred from the environment, such as the DirectionalHopper environment we will examine in Section 4.1, we add rewards $r_i$ to the set of inputs in Eqs. (7) and (8). In fact, any information that can be obtained from the environment could be added as an input to the loss function, e.g., exploration signals, the current timestep number, etc, and we leave further such extensions as future work.

To bootstrap the learning process, we add to $L_{\boldsymbol{\phi}}$ a guidance policy gradient signal $L_{\text{pg}}$ (in practice, we use the surrogate loss from PPO [26]), making the total loss

$$\hat{L}_{\boldsymbol{\phi}} = (1 - \alpha)L_{\boldsymbol{\phi}} + \alpha L_{\text{pg}}. \quad (9)$$

We anneal $\alpha$ from 1 to 0 over a finite number of outer-loop epochs. As such, learning is first derived mostly from the well-structured $L_{\text{pg}}$, while over time $L_{\boldsymbol{\phi}}$ takes over and drives learning completely after $\alpha$ has been annealed to 0.

## 4    Experiments

We apply our method to several randomized continuous control MuJoCo environments [1, 19, 4], namely RandomHopper and RandomWalker (with randomized gravity, friction, body mass, and link thickness), RandomReacher (with randomized link lengths), DirectionalHopper and DirectionalHalfCheetah (with randomized forward/backward reward function), GoalAnt (reward function based on the randomized target location), and Fetch (randomized target location). We describe these environments in detail in the supplementary materials. These environments are chosen because they require the agent to identify a randomly sampled environment at test time via exploratory behavior. Examples of the randomized Hopper environments are shown in Figure 9. The plots in this section show the mean value of 20 test-time training curves as a solid line, while the shaded area represents the interquartile range. The dotted lines plot 5 randomly sampled curves.

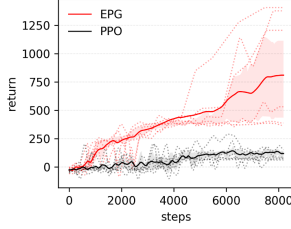

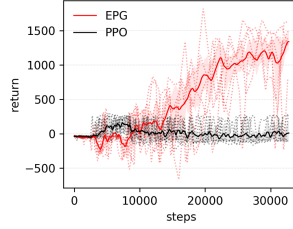

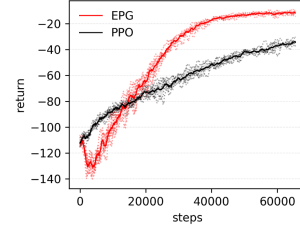

Figure 3: RandomHopper test-time training over 128 (policy updates) ×64 (update frequency) = 8196 timesteps: PPO vs no-reward EPG

Figure 4: RandomWalker test-time training over 256 (policy updates) ×128 (update frequency) = 32768 timesteps: PPO vs no-reward EPG

Figure 5: RandomReacher test-time training over 512 (policy updates) ×128 (update frequency) = 65536 timesteps: PG vs no-reward EPG.

## 4.1 Performance

We compare metatest-time learning performance, using the EPG loss function, against an off-the-shelf policy gradient method, PPO [26]. Figures 3, 4, 5, and 6 show learning curves for these two methods on the RandomHopper, RandomWalker, RandomReacher, and Fetch environments respectively at test time. The plots show the episodic return w.r.t. the number of environment steps taken so far. In all experiments, EPG agents learn more quickly and obtain higher returns compared to PPO agents.

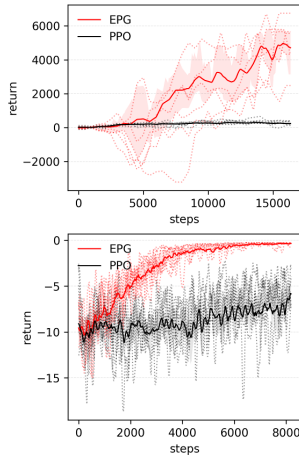

Figure 6: GoalAnt (top) and Fetch (bottom) environment learning over 512 and 256 (policy updates) ×32 (update frequency): PPO vs EPG (no reward for Fetch)

In these experiments, the PPO agent learns by observing reward signals whereas the EPG agent does not observe rewards (note that at test time, $\alpha$ in Eq. (9) equals 0). Observing rewards is not needed in EPG at metatest time, since any piece of information the agent encounters forms an input to the EPG loss function. As long as the agent can identify which task to solve within the distribution, it does not matter whether this identification is done through observations or rewards. This setting demonstrates the potential to use EPG when rewards are only available at metatraining time, for example, if a system were trained in simulation but deployed in the real world where reward signals are hard to measure.

Figures 7, 8, and 6 show experiments in which a signaling flag is required to identify the environment. Generally, this is done through a reward function or an observation flag, which is why EPG takes the reward as input in the case where the state space is partially-observed. Similarly to the previous experiments, EPG significantly outperforms PPO on the task distribution it is metatrained on. Specifically, in Figure 8, we compare EPG with both MAML (data from [6]) and RL² [5], finding that all three methods obtain similarly high performance after 8000 timesteps of experience. When comparing EPG to RL² (a method that learns a recurrent policy that does not reset the internal state upon trajectory resets), we see that RL² solves the DirectionalHalfCheetah task almost instantly through system identification. By learning both the algorithm and the policy initialization simultaneously, it is able to significantly outperform both MAML and EPG. However, this comes at the cost of generalization power, as we will discuss in Section 4.3.

## 4.2 Learning exploratory behavior

Without additional exploratory incentives, PG methods lead to suboptimal policies. To understand whether EPG is able to train agents that explore, we test our method and PPO on the DirectionalHopper and GoalAnt environments. In DirectionalHopper, each sampled Hopper environment either rewards the agent for forward or backward hopping. Note that without observing the reward, the agent cannot

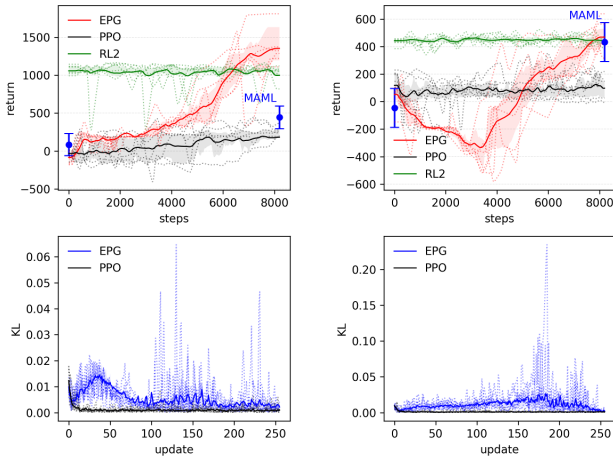
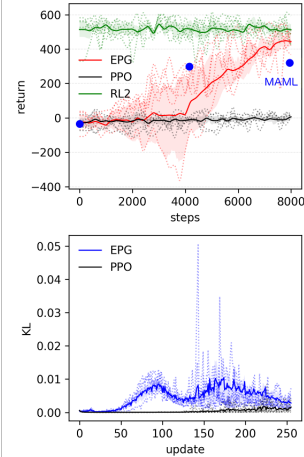

Figure 7: DirectionalHopper environment: each Hopper environment randomly decides whether to reward forward (left) or backward (right) hopping. In the right plot, we can see exploratory behavior, indicated by the negative spikes in the reward curve, where the agent first tries out walking forwards before realizing that backwards gives higher rewards.

Figure 8: DirectionalHalfCheetah environment from Finn et al. [6] (Fig. 5). Blue dots show 0, 1, and 2 gradient steps of MAML after metalearning a policy initialization.

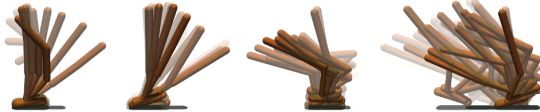

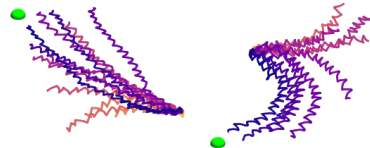

Figure 9: Example of learning to hop backward from a random policy in a DirectionalHopper environment. Left to right: sampled trajectories as learning progresses.

Figure 10: Sampled trajectories at test-time training on two GoalAnt environments: various directions are explored.

infer whether the Hopper environment desires forward or backward hopping. Thus we augment the environment reward to the input batches of the loss function in this setting.

Figure 7 shows learning curves of both PPO agents and agents trained with the learned loss in the DirectionalHopper environment. The learning curves give indication that the learned loss is able to train agents that exhibit exploratory behavior. We see that in most instances, PPO agents stagnate in learning, while agents trained with our learned loss manage to explore both forward and backward hopping and eventually hop in the correct direction. Figure 7 (right) demonstrates the qualitative behavior of our agent during learning and Figure 9 visualizes the exploratory behavior. We see that the hopper first explores one hopping direction before learning to hop backwards. The GoalAnt environment randomizes the location of the goal. Figure 10 demonstrates the exploratory behavior of a learning ant trained by EPG. The ant first explores in various directions, including the opposite direction of the target location. However, it quickly figures out in which quadrant to explore, before it fully learns the correct direction to walk in.

## 4.3 Generalization to out-of-distribution tasks

We evaluate generalization to out-of-distribution task learning on the GoalAnt environment. During metatraining, goals are randomly sampled on the positive x-axis (ant walking to the right) and at test time, we sample goals from the negative x-axis (ant walking to the left). Achieving generalization to the left side is not trivial, since it may be easy for a metalearner to overfit to the task metatraining distribution. Figure 11 (a) illustrates this generalization task. We compare the performance of EPG against MAML [6] and RL$^2$ [5]. Since PPO is not metatrained, there is no difference between both directions. Therefore, the performance of PPO is the same as shown in Figure 6.

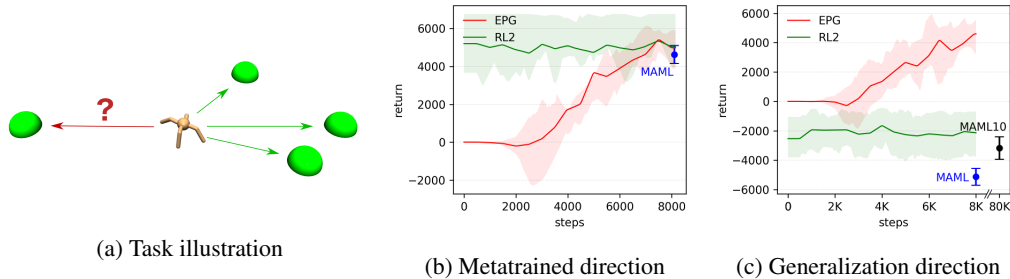

(a) Task illustration     (b) Metatrained direction     (c) Generalization direction

Figure 11: Generalization in GoalAnt: the ant has only been metatrained to reach targets on the positive x-axis (its right side). Can it generalize to targets on the negative x-axis (its left side)?

First, we evaluate all metalearning methods' performance when the test-time task is sampled from the training-time task distribution. Figure 11 (b) shows the test-time training curve of both $RL^2$ and EPG when the test-time goals are sampled from the positive x-axis. As expected, $RL^2$ solves this task extremely fast, since it couples both the learning algorithm and the policy. EPG performs very well on this task as well, learning an effective policy from scratch (random initialization) in 8192 steps, with final performance matching that of $RL^2$. MAML achieves approximately the same final performance after taking a single SGD step (based on 8000 sampled steps).

Next, we look at the generalization setting with test-time goals sampled from the negative x-axis in Figure 11 (c). $RL^2$ seems to have completely overfit to the task distribution, it has not succeeded in learning a general learning algorithm. Note that, although the $RL^2$ agent still walks in the wrong direction, it does so at a lower speed, indicating that it notices a deviation from the expected reward signal. When looking at MAML, we see that MAML has also overfit to the metatraining distribution, resulting in a walking speed in the wrong direction similar to the non-generalization setting. The plot also depicts the result of performing 10 gradient updates from the MAML initialization, denoted MAML10 (note that each gradient update uses a batch of 8000 steps). With multiple gradient steps, MAML does make progress toward improving the returns (unlike $RL^2$ and consistent with [7]), but still learns at a far slower rate than EPG. MAML can achieve this because it uses a standard PG learning algorithm to make progress beyond its initialization, and therefore enjoys the generalization property of generic PG methods.

In contrast, EPG evolves a loss function that trains agents to quickly reach goals sampled from negative x-axis, never seen during metatraining. This demonstrates rudimentary generalization properties, as may be expected from learning a loss function that is decoupled from the policy. Figure 10 shows trajectories sampled during the EPG learning process for this exact setup.

## 5 Discussion

We have demonstrated that EPG can learn a loss that is specialized to the task distribution it is metatrained on, resulting in faster test time learning on novel tasks sampled from this distribution. In a sense, this loss function internalizes an agent's notion of what it means to make progress on a task. In some cases, this eliminates the need for external, environmental rewards at metatest time, resulting in agents that learn entirely from intrinsic motivation [22].

Although EPG is trained to specialize to a task distribution, it also exhibits generalization properties that go beyond current metalearning methods such as $RL^2$ and MAML. Improving the generalization ability of EPG, as well other other metalearning algorithms, will be an important component of future work. Right now, we can train an EPG loss to be effective for one small family of tasks at a time, e.g., getting an ant to walk left and right. However, the EPG loss for this family of tasks is unlikely to be at all effective on a wildly different kind of task, like playing Space Invaders. In contrast, standard RL losses do have this level of generality – the same loss function can be used to learn a huge variety of skills. EPG gains on performance by losing on generality. There may be a long road ahead toward metalearning methods that both outperform standard RL methods and have the same level of generality.

## Footnotes

[1]More specifically, the average return over 3 sampled trajectories using the final policy for worker $w$.

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
