[Supplementary Material · Evolved_Policy_Gradients_Supp.pdf]

# Evolved Policy Gradients: Supplementary Materials

**Rein Houthooft**[*], **Richard Y. Chen**[*], **Phillip Isola**[*†×], **Bradly C. Stadie**[*†], **Filip Wolski**[*],
**Jonathan Ho**[*†], **Pieter Abbeel**[†]
OpenAI[*], UC Berkeley[†], MIT[×]

## A    Relation to Existing Literature

The concept of learning an algorithm for learning is quite general, and hence there exists a large body of somewhat disconnected literature on the topic.

To begin with, there are several relevant and recent publications in the metalearning literature [6, 5, 31, 14]. In [6], an algorithm named MAML is introduced. MAML treats the metalearning problem as in initialization problem. More specifically, MAML attempts to find a policy initialization from which only a minimal number of policy gradient steps are required to solve new tasks. This is accomplished by performing gradient descent on the original policy parameters with respect to the post policy update rewards. In Section 4.1 of Finn et al. [7], learning the MAML loss via gradient descent is proposed. Their loss has a more restricted formulation than EPG and relies on loss differentiability with respect to the objective function.

In a work concurrent with ours, Yu et al. [34] extended the model from [7] to incorporate a more elaborate learned loss function. The proposed loss involves temporal convolutions over trajectories of experience, similar to the method proposed in this paper. However, unlike our work, [34] primarily considers the problem of behavioral cloning. Typically, this means their method will require demonstrations, in contrast to our method which does not. Further, their outer objective does not require sequential reasoning and must be differentiable and their inner loop is a single SGD step. We have no such restrictions. Our outer objective is long horizon and non-differentiable and consequently our inner loop can run over tens of thousands of timesteps.

Another recent metalearning algorithm is $RL^2$ [5] (and related methods such as [31] and [14]). $RL^2$ is essentially a recurrent policy learning over a task distribution. The policy receives flags from the environment marking the end of episodes. Using these flags and simultaneously ingesting data for several different tasks, it learns how to compute gradient updates through its internal logic. $RL^2$ is limited by its decision to couple the policy and learning algorithm (using recurrency for both), whereas we decouple these components. Due to $RL^2$'s policy-gradient-based optimization procedure, we see that it does not directly optimize final policy performance nor exhibit exploration. Hence, extensions have been proposed such as E-$RL^2$ [28] in which the rewards of episodes sampled early in the learning process are deliberately set to zero to drive exploratory behavior.

Further research on meta reinforcement learning comprises a vast selection. The literature's vastness is further complicated by the fact that the research appears under many different headings. Specifically, there exist relevant literature on: life-long learning, learning to learn, continual learning, and multi-task learning. For example, [21, 20] consider self-modifying learning machines (genetic programs). If we consider a genetic program that itself modifies the learned genetic program, we can subsequently derive a meta-GP approach (See [28], for further discussion on how this method relates to the more recent metalearning literature discussed above). The method described above is sufficiently general that it encompass most modern metalearning approaches. For a further review of other metalearning approaches, see the review articles [24, 29, 30] and citation graph they generate.

There are several other avenues of related work that tackle slightly different problems. For instance, several methods attempt to learn a reward function to drive learning. EPG's outer loop depends on a learning signal, which is episode-wise cumulative reward instead of per-time-step-reward. Thus, unlike reward learning, EPG does not rely on explicit reward encoding for each time step, which could make it more general (e.g., EPG could use human feedback on each episode to judge the fitness of a learning session during metatraining). Christiano et al. [4] suggests learning from human feedback and the field of Inverse Reinforcement Learning [15] which recovers the reward from demonstrations. Both of these fields relate to our ideas on loss function learning. Similarly, [16, 17] apply population-based evolutionary algorithms to reward function learning in gridworld environments. This algorithm is encompassed by the algorithms we present in this paper. However, it is typically much easier since learning just the reward function is in many cases a trivial task (e.g., in learning to walk, mapping the observation of distance to a reward function). See also [25, 26] and [1] for additional evolutionary perspectives on reward learning. Other reward learning methods include the work of Guo et al. [8], which focuses on learning reward bonuses, and the work of Sorg et al. [27], which focuses on learning reward functions through gradient descent. These bonuses are typically designed to augment but not replace the learned reward and have not been shown to easily generalize across broad task distributions. Reward bonuses are closely linked to the idea of curiosity, in which an agent attempts to learn an internal reward signal to drive future exploration. Schmidhuber [19] was perhaps the first to examine the problem of intrinsic motivation in a metalearning context. The proposed algorithms make use of dynamic programming to explicitly partition experience into checkpoints. Further, there is usually little focus on metalearning the curiosity signal across several different tasks. Finally, the work of [9, 33, 2, 11, 13] studies metalearning over the optimization process in which metalearner makes explicit updates to a parametrized model in supervised settings.

Also worth mentioning is that approaches such as UVFA [18] and HER [3], which learn a universal goal-directed value function, somewhat resemble EPG in the sense that their critic could be interpreted as a sort of loss function that is learned according to a specific set of rules. Furthermore, in DDPG [12], the critic can be interpreted in a similar way since it also makes use of back-propagation through a learned function into a policy network.

## B    Test-time Algorithm Description

---
**Algorithm 2:** EPG test-time training
---
1  **[Input]**: learned loss function $L_\phi$ from EPG, MDP $\mathcal{M}$
2  Initialize buffer with $N$ zero tuples
3  Initialize policy parameter $\boldsymbol{\theta}$ randomly
4  **for** step $t = 1, \ldots, U$ **do**
5      Sample initial state $s_t \sim p_0$ if $\mathcal{M}$ needs to be reset
6      Sample action $a_t \sim \pi_{\boldsymbol{\theta}}(\cdot|s_t)$
7      Take action $a_t$ in $\mathcal{M}$, receive $r_t$, $s_{t+1}$, and termination flag $d_t$
8      Add tuple $(s_t, a_t, r_t, d_t)$ to buffer
9      **if** $t \bmod M = 0$ **then**
10         Calculate losses $L_i$ for steps $i = t - M, \ldots, t$ using buffer tuples $i - N, \ldots, i$
11         Sample minibatches mb from last $M$ steps shuffled, compute $L_{\mathrm{mb}} = \sum_{j \in \mathrm{mb}} L_j$, and
        update the policy parameter $\boldsymbol{\theta}$ and memory parameter (Eq. (5))
12 **[Output]**: A trained policy $\pi_{\boldsymbol{\theta}}$ for MDP $\mathcal{M}$
---

## C    Implementation details

In our experiments, the temporal convolutional layers of the loss function has 3 layers. The first layer has a kernel size of 8, stride of 7, and outputs 10 channels. The second layer has a kernel of 4, stride of 2, and outputs 10 channels. The third layer is fully-connected with 32 output units. Leaky ReLU activation is applied to each convolutional layer. The fully-connected component takes as input the trajectory features from the convolutional component concatenated with state, action, termination

signal, and policy output, as well as reward in experiments in which reward is observed. It has $1$ hidden layer with $16$ hidden units and leaky ReLU activation, followed by an output layer. The buffer size is $N \in \{512, 1024\}$. The agent's MLP policy has $2$ hidden layers of $64$ units with $\tanh$ activation. The memory unit is a 32-unit single layer with $\tanh$ activation.

We use $W = 256$ inner-loop workers in Algorithm 1, combined with $V = 64$ ES noise vectors. The loss function is evolved over 5000 epochs, with $\alpha$, as in Eq. (9) from the main text, annealed linearly from 1 to 0 over the first 500 epochs. The off-the-shelf PG algorithm (PPO) was moderately tuned to perform well on these tasks, however, it is important to keep in mind that these methods inherently have trouble optimizing when the number of samples drawn for each policy update batch is low. EPG's inner loop update frequency is set to $M \in \{32, 64, 128\}$ and the inner loop length is $U \in \{64 \times M, 128 \times M, 256 \times M, 512 \times M\}$. At every EPG inner loop update, the policy and memory parameters are updated by the learned loss function using shuffled minibatches of size 32 within each set of $M$ most recent transition steps in the replay buffer, going over each step exactly once. We tabulate the hyperparameters for each randomized environment in Table 1 in Section G.

Normalization according to a running mean and standard deviation were applied to the observations, actions, and rewards for each EPG inner loop worker independently (Algorithm 1) and for test-time training (Algorithm 2). Adam [10] is used for the EPG inner loop optimization and test-time training with $\beta_1 = 0.9$ and $\beta_2 = 0.999$, while the outer loop ES gradients are modified by Adam with $\beta_1 = 0$ and $\beta_2 = 0.999$ (which means momentum has been turned off) before updating the loss function. Furthermore, L2-regularization over the loss function parameters with coefficient $0.001$ is added to outer loop objective. The inner loop step size is fixed to $10^{-3}$, while the outer loop step size is annealed linearly from $10^{-2}$ to $10^{-3}$ over the first 2000 epochs.

## D  Environment Description

We describe the randomized environments used in our experiments in the following:

- *RandomHopper* and *RandomWalker*: randomized gravity, friction, body mass, and link thickness at metatraining time, using a forward-velocity reward. At test-time, the reward is not fed as an input to EPG.

- *RandomReacher*: randomized link lengths, using the negative distance as a reward at metatraining time. At test-time, the reward is not fed as an input to EPG, however, the target location is fed as an input observation.

- *DirectionalHopper* and *DirectionalHalfCheetah*[1]: randomized velocity reward function.

- *GoalAnt*: ant environment with randomized target location and randomized initial rotation of the ant. The velocity to the target is fed in as a reward. The target location is not observed.

- *Fetch*: randomized target location, the reward function is the negative distance to the target. The reward function is not an input to the EPG loss function, but the target location is.

## E  Additional Experiments

Two key components of Algorithm 1 are inner-loop training horizon $U$, and the agent's policy architecture $\pi_\theta$. In the next two paragraphs, we investigate metatest-time generalization to longer training horizons, and to different policy architectures:

**Longer training horizons**   We evaluate the effect of transferring to longer agent training periods at test time on the RandomHopper environment by increasing the test-time training steps $U$ in Algorithm 2 beyond the inner-loop training steps $U$ of Algorithm 1. Figure 1 (A) shows that the learning curve declines and eventually crashes past the train-time horizon, which demonstrates that Algorithm 1 has limited generalization beyond EPG's inner-loop training steps. However, we can overcome

Figure 1: Transferring EPG (metalearned using 128 policy updates on RandomHopper) to 1536 updates at test time: random policy init (A), init by sampled previous policies (B)

(a) 2 layers of 256 tanh units    (b) 2 layers of 64 ReLU units    (c) 4 layers of 64 tanh units

Figure 2: Transferring EPG (metalearned using 2-layer 64 tanh-unit policies on RandomWalker) to policies of unseen configurations at test time

this limitation by initializing each inner-loop policy with randomly sampled policies that have been obtained by inner-loop training in past epochs. Figure 1 (B) illustrates continued learning past the train-time horizon, validating that this modification effectively makes the learned loss function robust to longer training length at test time.

**Different policy architectures**  We evaluate EPG's transfer to different policy architectures by varying the number of hidden layers, the activation function, and hidden units of the agent's policy at test time (Algorithm 2), while keeping the agent's policy fixed at 2-layer with 64 tanh units during training time (Algorithm 1) on the RandomWalker environment. The test-time training curves on varied policy architectures are shown in Figure 2. Compared to the learning curve with the same train-time and test-time policy architecture (Figure 4 in the main text), the transfer performance is inferior. However, we still see that EPG produces a learned loss function that generalizes to policies other than it was trained on, achieving non-trivial walking behavior.

**Correlation with PPO updates**  The correlation between the gradients of our learned loss and the PPO objective is around $\rho = 0.5$ (Spearman's rank correlation coefficient) for the environments tested. This indicates that the gradients produced by the learned loss are related to, but different from, those produced by the PPO objective.

**Learning adaptive policy updates**  PG methods such as REINFORCE [32] suffer from unstable learning, such that a large learning step size leads to policy crashing during learning. To encourage smooth policy updates, methods such as TRPO [22] and PPO [23] were proposed to limit the distributional change from each policy update, through a hyperparameter constraining the KL-divergence between the policy distributions before and after each update. We demonstrate that EPG produces learned loss that adaptively scales the gradient updates.

With a learned loss function, the policy updates tend to shift the policy distribution less on each step, but sometimes produce sudden changes, indicated by the spikes. These spikes are highly noticeable in Figure 7, in which we plot individual test-time training curves for several randomized environments. The loss function has evolved in such a way to adapt its gradient magnitude to the current agent state: for example in the DirectionalHalfCheetah experiments, the agent first ramps up its velocity in one direction (visible by a increasing KL-divergence) until it realizes whether it is going in the

right/wrong direction. Then it either further ramps up the velocity through stronger gradients, or emits a turning signal via a strong gradient spike (e.g., visible by the spikes in Figure 7 (a) in column three).

In other experiments, we see a similar pattern: often the gradient will be small initially, and it gets increasingly larger the more environment information it has encountered.

**Learning without environment resets**   We show that it is straightforward to evolve a loss that is able to perform well on no-reset learning, such that the agent is never reset to a fixed starting location and configuration after each episode. Figure 3 shows the average return w.r.t. the epoch on the GoalAnt environment without reset. The ant continues learning from the location and configuration after each episode finishes and is reset to the starting point only when the target is reached. Qualitative inspection of the learned behavior shows that the agent learns how to reach the target multiple times during its lifetime. In comparison, running PPO in a no-reset environment is highly difficult, since the agent's policy tends to get stuck in a position it cannot escape from (leading to an almost flat zero-return learning curve). In some way, this demonstrates that EPG's learned loss guides the agent to avoid states from which it cannot escape.

Figure 3: The average return w.r.t. the epoch on the GoalAnt environment with no reset.

**Training performance w.r.t. evolution epoch**   Figure 4 shows the metatraining-time performance (calculated based on the noise-perturbed loss functions) w.r.t. the number ES epochs so far, averaged across 256 different inner-loop workers for various random seeds, on several of our environments. This experiment highlights the stability of finding well-performing loss function via evolution. All experiments use 256 workers over 64 noise vectors and 256 updates every 32 steps (8196-step inner loop).

**EPG loss input sensitivity**   In the reward-free case (e.g., RandomHopper, RandomWalker, RandomReacher, and Fetch), the EPG loss function takes four kinds of inputs: observations, actions, termination signals, and policy outputs, and evaluates entire buffer with $N$ transition steps. Which types of input and which time points in the buffer matter the most? In Figure 5, we plot the sensitivity of the learned loss function to each of these kinds of inputs by computing $||\frac{\partial L_{t=25}}{\partial x_t}||_2$ for different kinds of input $x_t$ at different time points $t$ in the input buffer. This analysis demonstrates that the loss is especially sensitive to experience at the current time step where it is being evaluated, but also depends on the entire temporal context in the input buffer. This suggests that the temporal convolutions are indeed making use of the agent's history (and future experience) to score the behavior.

**Effect of evolving policy initialization**   Prior works such as MAML [6] metalearn the policy initialization over a task distribution. While our proposed method, EPG, evolves the loss function parameters, we can also augment Algorithm 1 with simultaneously evolving the policy initialization in the ES outer loop. We investigate the benefits of evolving the policy initialization on top of EPG and PPO on our randomized environments. Figure 6 shows the comparison between EPG, EPG with evolved policy initialization (EPG+I), PPO, and PPO with evolved policy initialization (PPO+I). Evolving the policy initialization seems to help the most when the environments require little exploration, such as RandomWalker. However, the initialization plays a far less important role in DirectionalHalfCheetah and especially the GoalAnt environment. Hence the smaller performance difference between EPG and EPG+I.

(a) RandomHopper

(b) DirectionalHopper

(c) GoalAnt

(d) Fetch

Figure 4: Final returns averaged across 256 inner-loop workers w.r.t. the number outer-loop ES epochs so far in EPG training (Algorithm 1). We run EPG training on each environment across 5 different random seeds and plot the mean and standard deviation as a solid line and a shaded area respectively.

Figure 5: Loss input sensitivity: gradient magnitude of $L_{t=25}$ w.r.t. its inputs at different time steps within the input buffer. Notice not only the strong dependence on current time point ($t = 25$), but also the dependence on the entire buffer window.

Another interesting observation is that evolving the policy initialization, together with the EPG loss function (EPG+I), leads to qualitatively different behavior than PPO+I. In PPO+I, the initialization enables fast learning initially, before the return curves saturate. Obtaining a policy initialization that performs well without learning updates was impossible, since there is no single initialization that performs well for all tasks $\mathcal{M}$ sampled from the task distribution $p(\mathcal{M})$. In the EPG case however, we see that the return curves are often lower initially, but higher at the end of learning. By feeding the final return value as the objective function to the ES outer loop, the algorithm is able to avoid myopic return optimization. EPG+I sets the policy up for initial exploratory behavior which, although not beneficial in the short term, improves ultimate agent behavior.

**Individual test-time training curves** A detailed plot of how individual learners behave in each environment is shown in Figure 7. Looking at both the return and KL plots for the Directional-HalfCheetah and GoalAnt environments, we see that the agent ramps up its velocity, after which it either finds out it is going in the right direction or not. If it is going in the wrong direction initially, it provides a counter signal, turns, and then ramps up its velocity in the appropriate direction, increasing its return. This demonstrates the exploratory behavior that occurs in these environments. In the RandomHopper case, only a slight period of system identification exists, after which the velocity of the hopper is quickly ramped up (visible by the increasing KL divergences).

|  | (a) RandomWalker | (b) DirectionalHalfCheetah | (c) GoalAnt |
|---|---|---|---|

Figure 6: Effect of evolving the policy initialization (+I) on various randomized environments. test-time training curves with evolved policy initialization start at the same return value as those without evolved initialization. This is consistent with MAML trained on a wide task distribution (Figure 5 of [6]).

| Environment | workers $W$ | vectors $V$ | update freq. $M$ | updates | inner steps |
|---|---|---|---|---|---|
| RandomHopper | 256 | 64 | 64 | 128 | 8196 |
| RandomWalker | 256 | 64 | 128 | 256 | 32768 |
| RandomReacher | 256 | 64 | 128 | 512 | 65536 |
| DirectionalHopper | 256 | 64 | 64 | 128 | 8196 |
| DirectionalHalfCheetah | 256 | 64 | 32 | 256 | 8196 |
| GoalAnt | 256 | 64 | 32 | 512 | 16384 |
| Fetch | 256 | 64 | 32 | 256 | 8192 |

Table 1: EPG hyperparameters for different environments

# F    Computational efficiency

At meta-training time, EPG requires substantial computational resources. The number of sequential steps for each inner-loop worker in our algorithm is $E \times U$, using notation from Algorithm 1. In practice, this value may be very high, for example, each inner-loop worker takes approximately 196 million steps to evolve the loss function used in the RandomReacher experiments.

Improving computational efficiency is an important direction for future work. Currently, the outer-loop of EPG demands sequential learning. That is to say, one must first perform outer loop update $i$ before learning about update $i + 1$. This can bottleneck the metalearning cycle and create large computational demands. Finding ways to parallelize parts of this process, or increase sample efficiency, could greatly improve the practical applicability of our algorithm. Improvements in computational efficiency would also allow the investigation of more challenging tasks. Nevertheless, we feel the success on the environments we tested is non-trivial and provides a proof of concept of our method's power.

# G    Experiment Hyperparameters

The experiment hyperparameters for different environments are listed in Table 1.

## Footnotes

[1]Environment sourced from http://github.com/cbfinn/maml_rl.