[Reviews · NeurIPS 2018]

Reviewer 1



The authors present an approach for learning loss functions for reinforcement learning via a combination of evolutionary strategies as an outer loop and a simple policy gradient algorithm in the inner loop. Overall I found this to be a very interesting paper. My one criticism is that I would have liked to see a bit more of a study of what parts of the algorithm and the loss architecture are important. The algorithm itself is relatively simple. Although I appreciate the detail of Algorithm 1, to some degree I feel that this obscures the algorithm. In essense this approach corresponds to "use policy gradient in the inner-loop, and ES in the outer loop".More interesting is the structure of the loss architecture. It feels like this is giving an interesting inductive bias to the problem of learning the loss. It would be informative to see more of an ablation study of this architecture, i.e. if the loss were simplified to be a sum over "learned instantaneous rewards" and perhaps some adaptive discounting factor how much would the performance drop off. The bootstrapping process with PPO also seems quite important. But how is the performance affected without this bootstrapping? Does EPG fail to learn without this priming? The experiments are performed on a number of continuous control tasks and are quite thorough. I would have liked, however, to see more of an examination of the failure modes of EPG. It seems like there is no way to guarantee performance for this method when applied to problems outside of the training set, but I would have liked to see a more thorough examination of where this starts to fail. To some degree this is examined in the GoalAnt setting, but we don't see how far before this starts to break down. Minor comments: - In the introduction I'm not sure that the statement "RL methods optimize for short-term returns" is a fair statement. It's more that many current techniques may under-explore and not see more far-reaching rewards. I would go further in classifying "RL" as a problem rather than a set of solutions, under which the proposed EPG method is another "RL method". - line 82: typo "oftent".

Reviewer 2



The authors propose to meta-learn a differentiable parameterised loss function to be used in policy gradients. They show that this loss function generalizes better than RL^2 and MAML to out-of-training-distribution tasks, but it is outperformed by RL^2 in-distribution. The loss function is learned by a simple evolution strategy that performs random perturbations to the parameters, as used in [Salimans, Evolution strategies as a scalable alternative to reinforcement learning, 2017]. Quality: Overall I find the idea appealing. The idea and algorithm is conceptually straightforward. The loss network is quite complicated, it is hard to tell what the contributions of the components are and what it is actually doing. It would be useful to see an ablation study, for example on the ‘memory’ component, or the various components added to the context vector (Eqn (8)). The experiments showcase various properties of the algorithm. The evolved loss function (EPG) yields better solutions (at a fixed budget) than PPO. EPG is substantially outperformed by RL2 when the test task has the same distributions to the training task. However it outperforms RL2 in the case where there is a small distributional shift in the task e.g. the direction of a target is at a novel angle. I like the experiments showcasing the wins/losses compared to RL2. The comparison to PPO hides the training time for the EPG loss function. I did not see mention of how much additional training time EPG gets, or how PPO would perform given the same computational budget. Clarity: The paper is easy to read. The section detailing the architecture is too dense, I don’t think the algorithm could be replicated from this. However the more detailed Appendix, and open-sourced code help here. Originality: As far as I know, this approach to meta-learning in the context of deep-RL is novel. Significance: The authors demonstrate some nice generalization behaviour of the algorithm. Although it is not a clear winner over other meta-learning approaches, I think the originality of the approach is significant and could spark further work on learning parameterised loss functions.

Reviewer 3



The authors consider a meta learning framework where in the outer loop a parametrized loss function is learned from environment reward with a simple ES approach, and in the inner loop standard policy gradient optimization is performed with the (meta-)learned loss function. The method is demonstrated on two continuous control tasks with body randomization and results show that (during test time) using loss functions learned by the proposed framework, the agent is able to achieve better performance compared to PPO, without observing environment rewards. The idea of this paper is interesting from multiple perspectives: 1. Meta-learning useful and sharable priors for RL agent from a distribution of randomized tasks. 2. Reducing the reliance of environment rewards during test time (or deployment) 3. How to combine evolution training (ES) with RL, in a hierarchical way. The use of ES can be better motivated. A question might be: why not using RL with temporal abstraction (such as many Hierarchical RL papers) for the outer loop to learn the parameterized loss? ES may (or may not) be a better approach to RL since the optimization without BP is easier? And it can address longterm credit assignment problem better than RL with temporal abstraction? The authors claim that one contribution is the learned loss function reduces the reliance of external reward during testing time. This is also a very important message but without much support from experiments or relevant discussions. I think the main problem is the tasks are too simple, thus to me the results in this paper do not imply that this idea could work on real tasks (such sim2real transfer). This also raises the concern of if the loss function here is really a "loss", in a traditional sense, since the loss function here is actually a parameterized model with additional MLP and temporal convolutions, which is also "transferred" in the evaluation time. This could change the story from learning loss functions to reusing a pretrained hierarchical reward encoding module, which has been well studied before. This also explains why there's no need for external reward using the learned loss function -- the external reward could be already encoded in the additional parameters. One argument could be the evolved loss function can be generalizable to more diverse tasks while simple transfer learning by modulation would fail. Overall I think the paper has many interesting ideas even though the presentation could be further improved. I recommend acceptance.

Reviewer 4



This paper proposes to use evolutionary strategy based meta-learning approach to learn a loss function for gradient-based reinforcement learning tasks. Before policy optimization, this algorithm uses evolutionary strategy to optimize a temporal convolutional network which generates loss function to train policies over the distribution of environments. After the meta training, the agent using the learned loss function out-performs an agent using PPO. The authors also demonstrate the advantages of this approach comparing with other meta-learning algorithms such as MAML or RL^2. pros: - The paper is well-written and the figures are helpful for understanding. - No reward during test time; - Simple, and using evolutionary strategy as a tool for meta-learning with RL tasks opens new directions. - The proposed algorithm is well verified in the experiment section, and the details are explained in the appendix. cons: - Although the training process is parallelized, it still requires to use large samples to learn a loss function that generalizes. - It is not clear whether this method can be utilized for real-world robotic learning tasks. - The generalization ability of this method should be further investigated, such as how well the loss generalizes acrosses domains.